# Effect Modification of Vitamin D Supplementation by Histopathological Characteristics on Survival of Patients with Digestive Tract Cancer: Post Hoc Analysis of the AMATERASU Randomized Clinical Trial

**DOI:** 10.3390/nu11102547

**Published:** 2019-10-22

**Authors:** Hideyuki Yonaga, Shinya Okada, Taisuke Akutsu, Hironori Ohdaira, Yutaka Suzuki, Mitsuyoshi Urashima

**Affiliations:** 1Division of Molecular Epidemiology, The Jikei University School of Medicine, 3–25–8, Nishi-Shimbashi, Minato-Ku, Tokyo 105-8461, Japan; yonaga_h@jikei.ac.jp (H.Y.); taisuke0107.jusom@gmail.com (T.A.); 2Celgene K. K., JP TOWER 2–7–2 Marunouchi Chiyoda-ku, Tokyo 100-7010, Japan; 3Department of Pathology, International University of Health and Welfare Hospital, 537–3 Iguchi, Nasushiobara, Tochigi 329-2763, Japan; shinya1012@iuhw.ac.jp; 4Department of Surgery, International University of Health and Welfare Hospital, 537–3 Iguchi, Nasushiobara, Tochigi 329-2763, Japan; ohdaira@iuhw.ac.jp (H.O.); yutaka@iuhw.ac.jp (Y.S.)

**Keywords:** prognosis, survival, Vitamin D, supplementation, pathology

## Abstract

Some coauthors of this study previously performed the AMATERASU randomized, double-blind, placebo-controlled trial of postoperative oral vitamin D supplementation (2,000 IU/day) in 417 patients with stage I to III digestive tract cancer from the esophagus to the rectum who underwent curative surgery (UMIN000001977). We conducted a post-hoc analysis of the AMATERASU trial to explore the effects of modification of vitamin D supplementation by histopathological characteristics on survival. Among patients with poorly differentiated adenocarcinoma, the 5-year relapse-free survival rate of patients supplemented with vitamin D was 91% compared with 63% in the placebo group (hazard ratio [HR], 0.25; 95% confidence interval [CI], 0.08 to 0.78; *P* = 0.017; *P* for interaction = 0.023). Similarly, the 5-year overall survival rate was 92% in the vitamin D group compared with 72% in the placebo group (HR, 0.25; 95%CI, 0.07 to 0.94; *P* = 0.040; *P* for interaction = 0.012). In contrast, there were no significant effects in other histopathological characteristics between vitamin D and placebo groups. These findings generated the hypothesis that oral vitamin D supplementation may improve both relapse-free survival and overall survival in a subgroup of patients with poorly differentiated adenocarcinoma.

## 1. Introduction

Vitamin D is mainly obtained from skin exposed to sunlight, but is also available from diet or vitamin D supplementation. Vitamin D is metabolized in the liver to 25-hydroxyvitamin D (25[OH]D), and serum levels of this metabolite can be used as a biomarker of vitamin D status. The 25(OH)D is further activated in the kidneys by 1-alpha-hydroxylase to 1,25-dihydroxyvitamin D, which facilitates calcium absorption and plays an important role in bone health. A more recent discovery that most tissues, as well as cancers, have 1-alpha-hydroxylase to convert serum 25(OH)D to 1,25(OH)2D, along with vitamin D receptors, which are steroid hormone nuclear receptors that regulate a variety of genes within a cell [1], has provided new insights into the function of vitamin D. In particular, vitamin D has been demonstrated to induce differentiation of cancer cells in vitro [2,3,4], which leads to the hypothesis that vitamin D supplementation may improve prognosis of patients with poorly differentiated carcinoma in vivo.

Recent meta-analyses of observational studies showed that higher levels of serum 25(OH)D were associated with lower cancer mortality [5,6,7,8]. More recent meta-analyses of randomized clinical trials (RCTs) indicated that vitamin D supplementation reduced cancer mortality [9,10]. In contrast, three RCTs published separately in 2019 showed null results: Vitamin D supplementation did not improve progression-free survival of 139 patients with unresectable advanced or metastatic colorectal cancer (SUNSHINE) [11], relapse-free survival of 417 patients with resectable digestive tract cancer from the esophagus to the rectum (AMATERASU) [12], or cancer mortality of 25,871 participants as one of the secondary endpoints (VITAL) [13]. Discrepancies between the meta-analyses of observational studies [5,6,7,8] and the three recent RCTs [11,12,13] are probably caused by confounding, in that patients with higher serum 25(OH)D levels may have more healthy and active lifestyles [14,15,16]. In contrast, discrepancies between the meta-analyses of RCTs [9,10] and the three recent RCTs [11,12,13] are not clear. Moreover, all of the null trials showed significant effects of vitamin D supplementation on exploratory analyses, including adjustment for patient demographics, subgroup analyses of 25(OH)D levels at baseline, and analysis excluding the first 2 years of follow-up. These results suggest that vitamin D may not have significant effects on all patients with cancer, but rather may have significant effects on specific subgroups of these patients. Thus, further investigation to identify the specific subgroups for which vitamin D supplementation would be synergistically effective is worthwhile.

Vitamin D has been repeatedly shown to induce differentiation of immature cancer cells in vitro [2,3,4]. In contrast, to the best of our knowledge, vitamin D has not been assessed in terms of its effects by maturity levels of carcinoma, e.g., well, moderately, and poorly differentiated carcinoma through examining pathological specimen derived from patients. In this post-hoc analysis of the AMATERASU randomized trial (UMIN000001977), we used the pathological specimens to identify unique subgroup(s) of patients with cancer with certain histopathological characteristics, especially in terms of differentiated levels of carcinoma, in which vitamin D supplementation effectively improved or worsened relapse-free survival and overall survival.

## 2. Materials and Methods 

### 2.1. Study Design Overview

This study was performed as a post-hoc analysis using data from the AMATERASU trial (UMIN000001977), a randomized, double-blind, placebo-controlled trial that investigated the effect of postoperative vitamin D supplementation in patients with digestive tract cancer. The AMATERASU trial was conducted at the International University of Health and Welfare Hospital in Japan between January 2010 and February 2018. The ethics committee of the International University of Health and Welfare Hospital (ethic approval code: 13–B–263) as well as the Jikei University School of Medicine (ethic approval code: 21–216(6094)) approved the study protocol. Written, informed consent was obtained from all patients before surgery. Details of the study design are described in the original report [12]. The study included 417 patients with digestive tract cancer (esophageal cancer, 10%; gastric cancer, 42%; colorectal cancer, 48%) who underwent curative surgery. Eligible patients were randomized in a 3:2 ratio to receive vitamin D supplements (2,000 IU/day) (*n* = 251) or placebo (*n* = 166) and started the supplementation at the first outpatient visit after surgery. Efficacy outcome measures were relapse-free survival and overall survival. Patients were regularly examined in an outpatient setting to exclude cancer relapse, as required by the surgeon in charge. Neoadjuvant and/or adjuvant chemotherapies were administered to patients as appropriate based on tumor type and clinical stage. The median follow-up period was 3.5 years with a follow-up rate of 99.8%.

### 2.2. Study Population

Eligible patients in the AMATERASU trial were aged 30 to 90 years, had a histopathological diagnosis of stage I to III epithelial carcinoma of the digestive tract (esophagus, stomach, small intestine, colon, or rectum), underwent curative surgery with complete tumor resection, did not take vitamin D supplements or active vitamin D, and had no history of urinary tract stones. Patients were excluded if tumors were not resectable by surgery or serious postoperative complications occurred before starting study drug supplementation. This post-hoc study included patients who were randomized in the AMATERASU trial and for whom specimens were available for histopathological evaluation. Histopathological characteristics were retrospectively collected from pathological reports generated at the curative surgery at entry. Patients eligible for analyses were classified into subgroups based on histopathological characteristics: squamous cell carcinoma; tubular adenocarcinoma of well differentiated type (tub1); moderately differentiated type (tub2); poorly differentiated adenocarcinoma (por); signet-ring cell carcinoma (sig); mucinous adenocarcinoma (muc); papillary adenocarcinoma (pap); and others. As many patients had multiple histopathological components, histopathological subgroups were not mutually exclusive of each other.

### 2.3. Outcome Measures

The outcome measures were relapse-free survival and overall survival. The relapse-free survival was defined as elapsed time from the date of randomization (i.e., time from starting the study medication) to the earliest date of cancer relapse or death due to any cause. Overall survival was defined as elapsed time from the date of randomization to the date of death due to any cause.

### 2.4. Statistical Analysis

Dichotomous and continuous variables among the histopathological subgroups were compared by the chi-square test and the Kruskal-Wallis test, respectively. Paired and unpaired continuous variables were compared by the Wilcoxon signed-rank test and the Mann-Whitney test, respectively. Outcome measures were assessed according to the randomly assigned intervention, that is, vitamin D or placebo. The effects of vitamin D and placebo on risk of relapse or death and death were estimated using Nelson-Aalen cumulative hazard curves for outcomes. A Cox proportional hazards model was used to determine hazard ratios (HRs) and 95% confidence intervals (CIs). To explore whether vitamin D supplementation significantly affected outcomes by histopathological subgroup, P for interaction was analyzed based on a Cox regression model including three variables: 1) the assigned intervention (the vitamin D group); 2) the subgroup stratified by histopathological characteristics; and 3) the assigned intervention and the subgroup multiplied together as an interaction variable, by two-way interaction tests comparing all other patients excluding those with the focused histopathologic subgroup. Values with two-sided *P* < 0.05 were considered significant. All data were analyzed using Stata 14.0 (StataCorp LP, College Station, TX, USA).

## 3. Results

### 3.1. Study Population

From the 417 patients with digestive tract cancers randomly assigned to receive vitamin D supplements (*n* = 251) or placebo (*n* = 166), patients with an unknown histopathological subtype (11 patients receiving vitamin D supplements and 6 patients receiving placebo) were excluded from the analyses (Figure 1). Thus, histopathological characteristics were available for 400: 240 of 251 (96%) patients received vitamin D supplements and 160 of 166 (96%) patients received placebo. These 400 patients were included in the analyses.

### 3.2. Histopathological Characteristics

Five typical histopathological characteristics were observed in the 400 patients: well differentiated (*n* = 218), moderately differentiated (*n* = 166), poorly differentiated (*n* = 81), signet-ring cell carcinoma (*n* = 44), and squamous cell carcinoma (*n* = 32) (Table 1). Patients who had two or more characteristics were counted in each category. Because of the small number of patients with mucinous carcinoma (*n* = 26), papillary carcinoma (*n* = 16), and others, that is, neuroendocrine tumor, adenoid cystic carcinoma, and unclassified (*n* = 6), further analyses for these histopathological characteristics were not conducted.

### 3.3. Patient Demographics

Baseline patient demographics in subgroups stratified by histopathological characteristics are shown in Table 1. The cancer site was significantly different in all histopathological subgroups when comparing the absence or presence of each histopathological characteristic (e.g., well differentiated vs. non-well differentiated, moderately differentiated vs. non-moderately differentiated, etc.). Both well-differentiated and moderately differentiated subgroups mainly consisted of patients with gastric cancer and patients with colorectal cancer. The poorly differentiated and signet-ring cell carcinoma subgroups mostly included patients with gastric cancer (91%). The squamous cell carcinoma subgroup was only documented in patients with esophageal cancer. Patients with the poorly differentiated and signet-ring cell carcinoma were most likely to have stage I disease, whereas those with squamous cell carcinoma were most likely to have stage II or stage III disease. In the signet-ring cell carcinoma subgroup, more patients received placebo, by chance. Serum levels of 25(OH)D at baseline (subgroup and median) were lower in the squamous cell carcinoma subgroup. Most patients in the squamous cell carcinoma subgroup were men. Patients in the well-differentiated subgroup were older, whereas those in the signet-ring group were younger. Body mass index was lower in patients in the squamous cell carcinoma subgroup compared with other subgroups. Among comorbidities, the prevalence of hypertension was lower in patients in the signet-ring cell carcinoma subgroup; the prevalences of diabetes mellitus and endocrine disease were lower in patients in the squamous cell carcinoma subgroup; and the prevalence of cardiovascular disease was higher in patients in the well-differentiated subgroup. Adjuvant chemotherapy was performed less frequently in patients in the well-differentiated subgroup and more frequently in patients in the squamous cell carcinoma subgroup. Patients’ characteristics between vitamin D and placebo were not significantly different in each histopathological subgroup except age in the subgroup of signet-ring cell carcinoma.

### 3.4. Effect Modification of Vitamin D Supplementation on Survival by Histopathological Subgroup

Table 2 shows results of the analyses for risk of relapse or death and of death in the histopathological subgroups. Significant beneficial effects by vitamin D supplementation were found in the poorly differentiated subgroup on both risk of relapse or death and risk of death. Although findings were not significant, similar point estimates of vitamin D effect on risk of relapse or death and risk of death were observed in the signet-ring cell carcinoma subgroup. The effects of vitamin D in the well differentiated, moderately differentiated, and squamous cell carcinoma subgroups were also not significant.

### 3.5. Effect Modification of Vitamin D Supplementation in the Poorly Differentiated Histopathological Subgroup 

To further address the relationship between vitamin D supplementation and poorly differentiated adenocarcinoma, cumulative hazard curves for risk of relapse or death and for risk of death were compared between vitamin D or placebo in both the poorly differentiated subgroup and the non-poorly differentiated subgroup in which the patients did not have poorly differentiated adenocarcinoma (Figure 2). The hazard curve for relapse or death (HR, 0.25; 95%CI, 0.08–0.78; *P* = 0.017) (Figure 2A) and the hazard curve for death (HR, 0.25; 95%CI, 0.07–0.94; *P* = 0.040) (Figure 2C) in the poorly differentiated subgroups were significantly improved in the vitamin D group compared with placebo group. The 5-year relapse-free survival and the 5-year overall survival in the poorly differentiated subgroups were 91% and 92% in the vitamin D group, respectively, versus 63% and 72% in the placebo group, respectively. On the other hand, in the non-poorly differentiated subgroup, the hazard curve for relapse or death (HR, 1.02; 95%CI, 0.64–1.61; *P* = 0.95) (Figure 2B) and the hazard curve for death (HR, 1.51; 95%CI, 0.82–2.78; *P* = 0.18) (Figure 2D) were not significant. The 5-year relapse-free survival and the 5-year overall survival in the non-poorly differentiated subgroup were 72% and 78% in the vitamin D group, respectively, compared with 71% and 81% in the placebo group, respectively. There was a significant two-way interaction between in the poorly differentiated and non-poorly differentiated subgroups for both risk of relapse or death (*P* = 0.023 for interaction) (Figure 2A and Figure 2B) and risk of death (*P* = 0.012 for interaction) (Figure 2C and Figure 2D). 

## 4. Discussion

In this post-hoc analysis of the AMATERASU trial, postoperative oral vitamin D supplementation (2,000 IU/day) had significant benefits on relapse-free survival and overall survival in the subgroup of patients with digestive tract cancer with poorly differentiated adenocarcinoma, but not in any other subgroup based on histopathological characteristics. In 2018, 9.6 million persons died from cancer worldwide, of which more than 2.1 million (22.5%) were from digestive tract cancers from the esophagus to the rectum [17]. In this study, poorly differentiated cancer was seen in 20% of patients, in whom vitamin D supplementation reduced 5-year relapse-free survival and overall survival by approximately 30% compared with placebo (91% vs. 63%) and 20% (92% vs. 72%), respectively. Even if vitamin D supplementation had a positive impact on only a portion (i.e., 20%) of patients included in this study, because the number of patients who die from digestive tract cancer is so high worldwide, vitamin D may have the potential to decrease many relapses and save lives. Moreover, vitamin D is a natural compound that side effects are considered to be rare. In contrast, patients with cancer treated by immune checkpoint inhibitors, which have been increasingly used in cancer therapy, frequently experience adverse events [18,19,20]. In particular, in terms of fatal toxic events, worldwide from 2009 through January 2018, 613 patients with cancer died due to treatments with combinations of immune checkpoint inhibitors [21]. Moreover, vitamin D is less expensive than other therapies. By simply being exposed to sunlight (e.g., exercising outside), patients can increase their 25(OH)D levels at no cost. 

Although there were no significant effects, a trend toward significant effects on survival was observed in the signet-ring cell carcinoma subgroup, which is classified as an undifferentiated type, as well as in the poorly differentiated adenocarcinoma subgroup. Therefore, it is possible that vitamin D may improve survival time in patients with undifferentiated, but not differentiated carcinoma. In-vitro experiments [2,3,4] have shown the potential for vitamin D to induce differentiation of undifferentiated cancer cells in vivo. In fact, a clinical pilot trial indicated that vitamin D may increase differentiation in the normal colorectal mucosa of patients with colorectal adenoma [22], although that trial did not include patients with cancer. 

There are several limitations to this study. First, this study was performed as a post-hoc subgroup analysis of the AMATERASU clinical trial. Although histopathological data were prospectively collected as adenocarcinoma, squamous cell carcinoma, or others, information of more detailed subgroups such as poorly differentiated used in this study were retrospectively retrieved from reports, each of which was judged by a single pathologist. Thus, misclassifications were possible in classifying subgroups. These misclassifications might have occurred with equal frequency in the vitamin D and placebo groups, which could have biased results toward the null hypothesis. Second, sample size was not calculated for the subgroup analyses. In addition, subgroup analyses may increase the probability of a type I error due to multiple comparisons. Thus, these findings must be considered exploratory and interpreted with caution.

## 5. Conclusions

These results generated a hypothesis that in the subgroup of patients with digestive tract cancer with poorly differentiated adenocarcinoma, vitamin D supplementation, compared with placebo, may improve relapse-free survival as well as overall survival.

## Figures and Tables

**Figure 1 nutrients-11-02547-f001:**
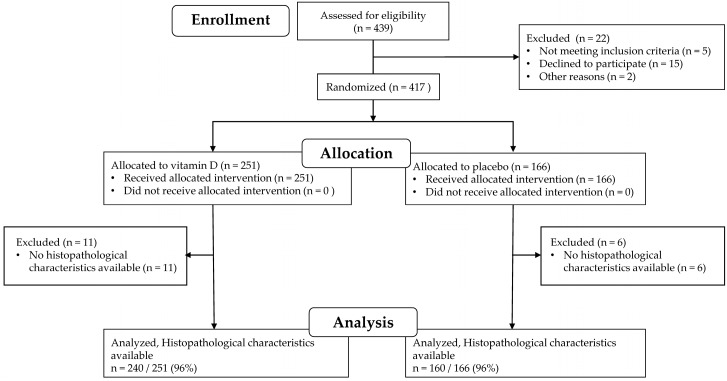
Patient flow through the AMATERASU trial and this post-hoc analysis.

**Figure 2 nutrients-11-02547-f002:**
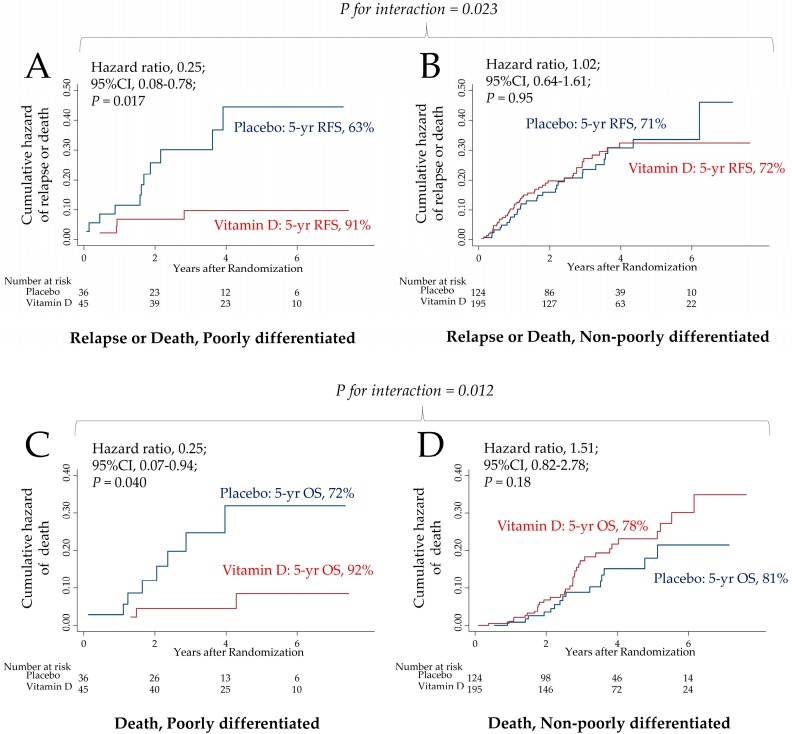
Effect of vitamin D supplementation on risk of relapse or death and risk of death: Poorly differentiated vs. non-poorly differentiated. Cumulative hazard curves between vitamin D or placebo were compared (**A**) for relapse or death in the poorly differentiated subgroup, (**B**) for relapse or death in the non-poorly differentiated subgroup, (**C**) for death in the poorly differentiated subgroup, and (**D**) for death in the non-poorly differentiated subgroup. P value was determined using a Cox proportional hazards model. P for interaction between the assigned intervention (vitamin D or placebo) and the histopathological subgroups (poorly differentiated or non-poorly differentiated) was tested based on a Cox regression model. CI, confidence interval.

**Table 1 nutrients-11-02547-t001:** Patient demographics in subgroups stratified by histopathological characteristics *.

Histopathological Characteristics, n (%)	Well 218 (55)	Moderately 166 (42)	Poorly 81 (20)	Signet-Ring 44 (11)	SCC 32 (8.0)
Site of cancer, n (%)					
Esophageal	2 (0.9)	4 (2.4)	0 (0.0)	0 (0.0)	32 (100)
Gastric	71 (33)	85 (51)	74 (91)	42 (95)	0 (0.0)
Small intestine	2 (0.9)	0 (0.0)	0 (0.0)	0 (0.0)	0 (0.0)
Colorectal	143 (66)	77 (46)	7 (8.6)	2 (4.6)	0 (0.0)
*P*-value	<0.001	<0.001	<0.001	<0.001	<0.001
Cancer stage, n (%)					
I	99 (45)	63 (38)	48 (59)	29 (66)	4 (13)
II	56 (26)	49 (30)	18 (22)	7 (16)	15 (47)
III	63 (29)	54 (33)	15 (19)	8 (18)	13 (41)
*P*-value	0.31	0.38	0.002	0.003	0.001
Intervention, n (%)					
Vitamin D	139 (64)	97 (58)	45 (56)	20 (45)	19 (59)
Placebo	79 (36)	69 (42)	36 (44)	24 (55)	13 (41)
*P*-value	0.093	0.59	0.36	0.037	0.94
Subgroup of 25(OH)D, n (%)					
Low: <20 ng/mL	85 (40)	66 (40)	35 (44)	16 (36)	20 (65)
Middle: ≥ 20 and ≤ 40 ng/mL	125 (58)	95 (58)	45 (56)	28 (64)	11 (35)
High: >40 ng/mL	4 (1.9)	2 (1.2)	0 (0.0)	0 (0.0)	0 (0.0)
*P*-value	0.23	0.71	0.52	0.44	0.037
25(OH)D, median (IQR), ng/mL	22 (16–27)	21 (16–27)	20.5 (15–26)	22 (16.5–26)	17 (13–22)
*P*-value	0.13	0.96	0.36	0.66	0.012
Sex, n (%)					
Male	144 (66)	115 (69)	53 (65)	26 (59)	29 (91)
Female	74 (34)	51 (31)	28 (35)	18 (41)	3 (9.4)
*P*-value	0.84	0.32	0.82	0.27	0.003
Age quartile, n (%)					
Q1, 35–59 y	46 (21)	44 (27)	27 (33)	19 (43)	5 (16)
Q2, 60–65 y	39 (18)	37 (22)	22 (27)	12 (27)	10 (31)
Q3, 66–73 y	65 (30)	43 (26)	16 (20)	8 (18)	6 (19)
Q4, 74–90 y	68 (31)	42 (25)	16 (20)	5 (11)	11 (34)
*P*-value	0.004	0.85	0.050	0.004	0.26
BMI quartile, n (%)					
Q1, 15.0–19.7 kg/m^2^	46 (21)	34 (21)	11 (14)	11 (25)	19 (61)
Q2, 19.8–21.8 kg/m^2^	55 (25)	45 (27)	24 (30)	12 (27)	9 (29)
Q3, 21.9–23.7 kg/m^2^	54 (25)	40 (24)	23 (28)	8 (18)	3 (9.7)
Q4, 23.8–37.3 kg/m^2^	63 (29)	46 (28)	23 (28)	13 (30)	0 (0.0)
*P*-value	0.21	0.40	0.091	0.71	<0.001
History of other cancers, n (%)	6 (2.8)	7 (4.2)	3 (3.7)	2 (4.6)	0 (0.0)
*P*-value	0.37	0.51	0.91	0.69	0.26
Comorbidities, n (%)					
Hypertension	92 (42)	60 (36)	27 (33)	10 (23)	10 (31)
*P*-value	0.096	0.42	0.29	0.023	0.38
Diabetes mellitus	47 (22)	24 (14)	10 (12)	6 (14)	0 (0.0)
*P*-value	0.003	0.35	0.26	0.59	0.009
Endocrine disease	29 (13)	22 (13)	10 (12)	6 (14)	0 (0.0)
*P*-value	0.60	0.70	0.96	0.81	0.026
Cardiovascular disease	22 (10)	12 (7.2)	6 (7.4)	2 (4.6)	1 (3.1)
*P*-value	0.031	0.86	0.97	0.43	0.33
Chronic kidney disease	3 (1.4)	2 (1.2)	0 (0.0)	0 (0.0)	0 (0.0)
*P*-value	0.82	0.68	0.21	0.39	0.47
Asthma	3 (1.4)	2 (1.2)	0 (0.0)	0 (0.0)	0 (0.0)
*P*-value	0.11	0.38	0.38	0.54	0.61
Orthopedic disease	1 (0.5)	0 (0.0)	1 (1.2)	0 (0.0)	0 (0.0)
*P*-value	0.90	0.23	0.29	0.62	0.68
Adjuvant chemotherapy, n (%)	64 (29)	64 (39)	25 (31)	15 (34)	18 (56)
*P*-value	0.001	0.47	0.24	0.73	0.016

* As many patients had multiple histopathological components, histopathological subgroups were not mutually exclusive of each other. Percentages may not equal 100% due to rounding. *P*-value was calculated by the chi-square test or the Kruskal-Wallis rank test. 25(OH)D, 25-hydroxyvitamin D; BMI, body mass index (weight [kg]/height squared [m^2^]); IQR, interquartile range; Q1, first quartile; Q2, second quartile; Q3, third quartile; Q4, fourth quartile; Well, well differentiated type of tubular adenocarcinoma; Moderately, moderately differentiated type of tubular adenocarcinoma; Poorly, poorly differentiated adenocarcinoma; Signet-ring, signet-ring cell carcinoma; SCC, squamous cell carcinoma.

**Table 2 nutrients-11-02547-t002:** Effect of Vitamin D supplementation on risk of relapse or death and death in subgroups stratified by histopathological characteristics.

		Risk of Relapse or Death				Risk of Death			
Subgroups	N	Events	HR	95%CI	*p*-Value	Events	HR	95%CI	*p*-Value
Well									
Vitamin D	139	28	0.81	0.46–1.44	0.48	19	0.82	0.40–1.65	0.57
Placebo	79	20	1.00			13	1.00		
Moderately									
Vitamin D	97	22	1.06	0.56–2.02	0.86	15	1.31	0.57–2.99	0.53
Placebo	69	16	1.00			9			
Poorly									
Vitamin D	45	4	0.25	0.08–0.78	0.017	3	0.25	0.07–0.94	0.040
Placebo	36	11	1.00			8	1.00		
Signet-ring									
Vitamin D	20	1	0.19	0.02–1.57	0.12	1	0.30	0.03–2.65	0.28
Placebo	24	6	1.00			4	1.00		
SCC									
Vitamin D	19	8	0.94	0.33–2.73	0.91	7	1.39	0.35–5.49	0.64
Placebo	13	6	1.00			3	1.00		

*P* value was determined using a Cox proportional hazards model. CI, confidential interval; HR, hazard ratio; Well, well differentiated type of tubular adenocarcinoma; Moderately, moderately differentiated type of tubular adenocarcinoma; Poorly, poorly differentiated adenocarcinoma; Signet-ring, signet-ring cell carcinoma; SCC, squamous cell carcinoma.

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
