# Peer review of "Effect Modification of Vitamin D Supplementation by Histopathological Characteristics on Survival of Patients with Digestive Tract Cancer: Post Hoc Analysis of the AMATERASU Randomized Clinical Trial"

_nutrients, 2019, doi:10.3390/nu11102547_

Round 1
Reviewer 1 Report
The authors have tested an interesting idea and the manuscript is well written. However few clarifications need to be made in the statistical analysis.
Line 109: Did authors separate the effects of patients’ sex, BMI and age assigned to receive vitamin D? If not, what was the justification for not considering those variables, as it would be interesting to understand whether different genders with different BMI and ages respond to vitamin D differently.
Reviewer 2 Report
Yonaga et al provide interesting post hoc findings in regard to the effect of vitamin D supplementation in patients with poorly differentiated GI cancer. I only have minor comments and questions:
The post hoc analysis provide data divided by histopathology, but nothing is available in the comparison between vitamin D and placebo supplemented subjects stratified by histopathology. This is probably due to the small n number, but I feel that it should be tentatively addressed, as significant difference in these subgroups (age/BMI/other comorbidities) may hinder data.
Moreover, in the discussion section, it is said: Even purchasing vitamin D supplements is still less expensive than immune checkpoint inhibitors.
This is obviously logic, but I believe it implies that vitamin D could substitute immune checkpoint inhibitors in treating oncologic patients, a statement mostly utopian given available evidence. I would rephrase this.
